# The coevolution of lifespan and reversible plasticity

Irja I. Ratikainen [1,2] & Hanna Kokko [3]

Reversible phenotypic plasticity, the ability to change one's phenotype repeatedly throughout life, can be selected for in environments that do not stay constant throughout an individual's lifetime. It might also mitigate senescence, as the mismatch between the environment and a non-plastic individual's traits is likely to increase as time passes. To understand why reversible plasticity may covary with lifespan, studies tend to assume unidirectional causality: plasticity evolves under suitable rates of environmental variation with respect to life history. Here we show that if lifespan also evolves in response to plasticity, then long life is not merely a context that sets the stage for lifelong plasticity. Instead, the causality is bidirectional because plasticity itself can select for longevity. Highly autocorrelated environmental fluctuations predict low investment in reversible plasticity and a phenotype that is poorly matched to the environment at older ages. Such environments select for high reproductive effort and short lifespans.

[1] Centre for Biodiversity Dynamics, Department of Biology, Faculty of Natural Sciences and Technology, Norwegian University of Science and Technology (NTNU), Høgskoleringen 5, NO-7491 Trondheim, Norway. [2] Institute of Biodiversity, Animal Health and Comparative Medicine, Graham Kerr Building, University of Glasgow, Glasgow G12 8QQ, UK. [3] Department of Evolutionary Biology and Environmental Studies, University of Zurich, Winterthurerstrasse 190, 8057 Zurich, Switzerland. Correspondence and requests for materials should be addressed to I.I.R. (email: irja.i.ratikainen@ntnu.no)

Phenotypic plasticity is defined as the ability of one genotype to produce more than one phenotype depending on some environmental variable[1]. This definition differs from phenotypic switching[2], a phenomenon where phenotypic change occurs without being related to the state of the environment as experienced by the organism. Organisms clearly vary in the extent of plasticity[3], the age in which plasticity occurs[4], and the general tendency for plasticity to be age dependent[5]. When the environment varies at a suitable rate with respect to the individual's life history, we expect phenotypic plasticity[6,7] as opposed to displaying a rigid expression of any genotype; this can extend to lifelong plasticity via reversible morphological change[8] or behavioural changes[9].

Models generally suggest that organisms that are long lived relative to the rate of environmental fluctuations should be more plastic[10,11]. Empirically, there appears to be support for the idea that lifespan or slow life histories correlate with large brains or higher learning rates (which presumably covary positively with behavioural flexibility) in mammals[12], birds[13,14], and butterflies[15]. The direction of causality is subject to debate[16]. Is a long life a context that simply sets the stage for lifelong plasticity, or can reversible plasticity itself select for increased longevity? Theoretical studies tend to take the lifespan as the driver of patterns, and ask whether plasticity then evolves[17–19]. The possibility that plasticity itself might drive life history evolution is understudied, even though there are hints from (non-plastic) life history theory that old individuals show more mismatch with the environment[20].

Here we complement insights from age-dependent plasticity[5] and the age-dependent expectation of phenotypic mismatch with the environment leading to senescence[20] to predict how the life history of a species is expected to coevolve with plasticity. We allow phenotypes to be updated according to evolving schedules. The modification of the phenotype usually occurs in a direction that improves the match between the environment and the phenotype, but the updating process is error-prone and we also investigate how the magnitude of error changes evolutionary expectations. We assume that each phenotypic updating event is costly, compared with leaving one's phenotype unchanged. The mismatch between the phenotype and the environment increases with the time since the last update (leading to a form of

senescence). This increase should select for higher reproductive effort in early life at the cost of survival—i.e. a 'fast' life history—in organisms that do not update their phenotypes as time progresses. If, on the other hand, an organism repeatedly updates its phenotype, it is less likely to senesce due to being out of date. This selects for a 'slow' life history where the emphasis is on reaping the benefits of a potentially long life. Given the costs of updating, a recent update might be best followed by a period of no updating, leading to an evolutionary question of the schedule according to which an organism should perform new updates.

## Results

**Model summary.** Our individual-based simulation follows genotypes, phenotypes, survival, and reproduction individually for all members of the population. Individual lives last several time steps without a strict upper bound, thus generations overlap. The environment, $E$, changes from one time step to the next, with the parameter $p$ determining the autocorrelation in the environmental states. The environmental state is operationally defined by the value of the phenotype that is optimal for this environment. An individual with a phenotype that deviates from this has nonzero mismatch $m_{i,t}$ with negative impact on current reproduction as well as survival to the next time step. Survival simultaneously trades off with reproductive effort.

Each individual has a two-locus genotype, where the $u$ locus determines an individual's update schedule and $r$ its reproductive effort. Each individual, in each time step, either updates or does not update its phenotype, with the update probability modelled as an increasing function of the time $T_u$ since the last update, with $u$ determining how steeply the probability rises with $T_u$ (high $u$ implies frequent updating, Fig. 1a). We assume some plasticity always occurs, in that newborn individuals (whose $T_u$ is undefined) perform an update with probability 1 regardless of $u$. Updating does not completely reset mismatch to zero but to a stochastic value that increases with an error term $\varepsilon$. This term can refer to, e.g., inappropriate phenotypic responses to the current environment (due to, for example, evolutionary lags), or to an organisms's inability to measure the state of the environment perfectly.

In each time step, all individuals can reproduce, with those that updated their phenotype paying a cost of updating (while

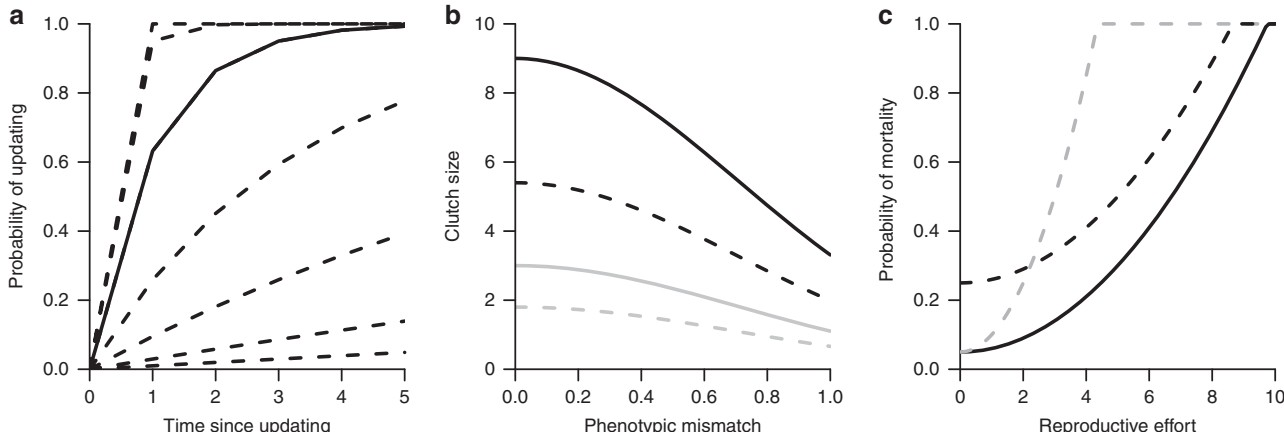

**Fig. 1** Model assumptions. **a** Probability of updating increases with time since last updating, but the rate of increase is regulated by $u$. Solid line is for $u = 1$, dashed lines above are for larger values of $u = \{3, 10\}$ and dashed lines below are for lower values of $u = \{0.3, 0.1, 0.03, 0.01\}$. **b** Clutch size, $c$, decreases with phenotypic mismatch, $m$, with the environment. Black lines are for high reproductive effort $r = 9$, grey indicates lower reproductive effort $r = 3$. Solid lines indicate clutch size when the individual did not update their phenotype, while dashed lines indicate the lower clutch size when individuals update due to costs of updating, $\kappa = 0.4$. **c** Mortality probability increases with reproductive effort, $r$. Solid line shows the probability of mortality when the individual has no phenotypic mismatch and $\rho = 0.01$, the black dashed line shows the mortality for individuals that mismatch their environment by 0.2 and the grey dashed line indicates mortality for when cost of reproduction, $\rho$, is increased to 0.05 (see Supplementary Note 2)

simultaneously potentially improving net performance due to reduced mismatch with the environment). Clutch size $c_{i,t}$ depends on the parent's investment in reproduction, its mismatch, and whether it performed an update (thus paying updating costs) (Fig. 1b). Offspring inherit the parent's $u_i$ (for updating schedule) and $r_i$ (for reproductive effort) with some small probability of mutation. Parental survival depends on mismatch and a cost of reproductive effort (which depends on $r_i$) (Fig. 1c). Dead parents create vacancies that can be filled by recruits; juveniles that do not recruit are removed. The simulations were run for 50,000 time steps, $t_{max}$, and after this we recorded the genotypes of all the individuals and their age.

We vary both the autocorrelation of the environment, $p$, and the updating error, $\varepsilon$. For each combination of $p$ and $\varepsilon$ (termed

'environmental scenario') we start the simulation from a range of genetic values for $u$ and $r$. The model is described in detail in the Methods section.

**Coevolution of reversible plasticity and lifespan.** Our results show strong support for the idea that reversible plasticity and lifespan can coevolve. We visualize these effects by presenting clouds of phenotypes (strategies) that differ in the environments they evolved in (effectively an interspecific or interpopulation comparison). Species may differ in environmental autocorrelations ($p$, indicated with colour in Fig. 2) and/or the magnitude of the updating error ($\varepsilon$, indicated with symbol shape). Repeated runs with the same parameters typically lead to similar levels of

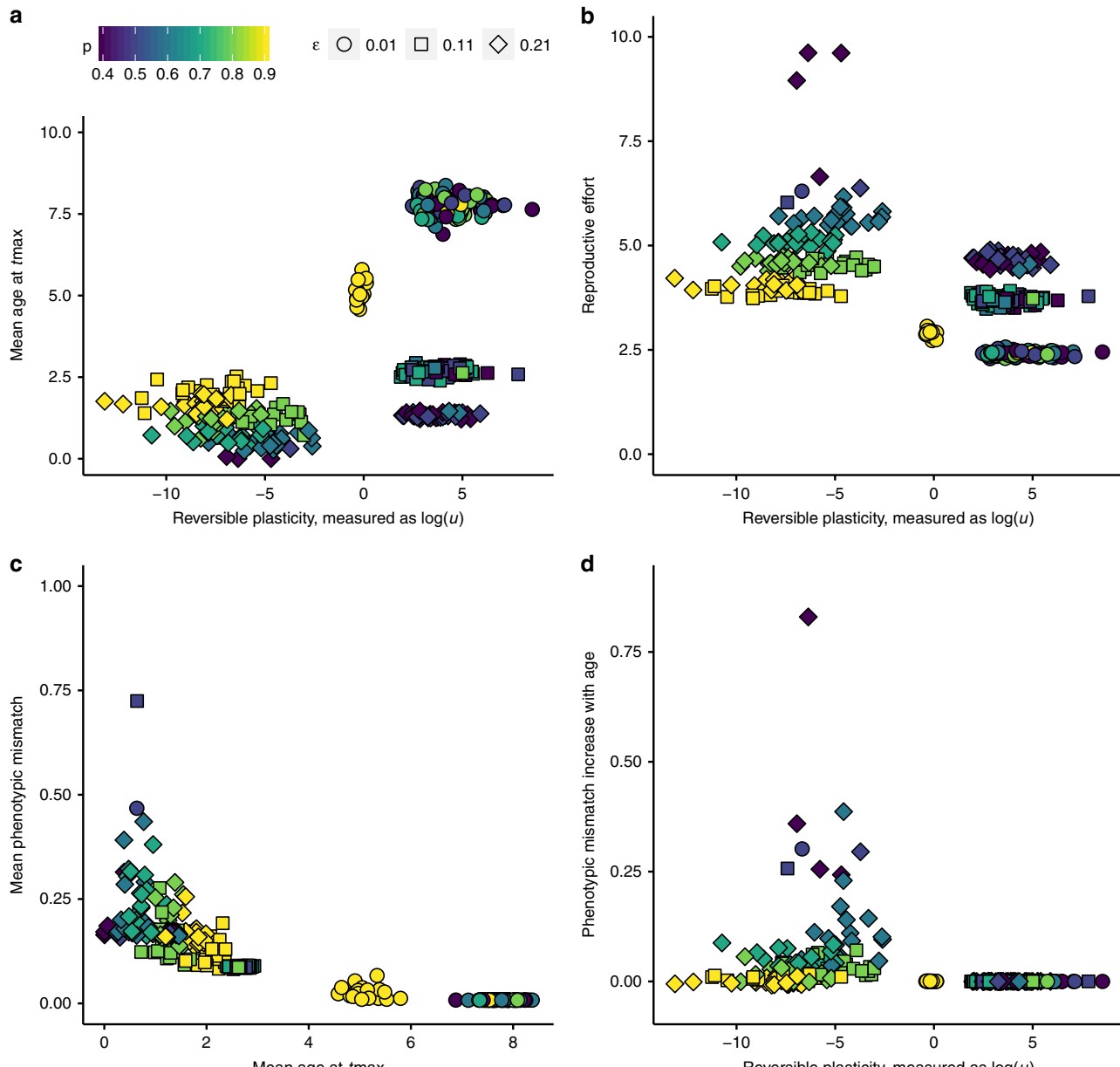

**Fig. 2** Relationships between populations mean trait values at the end of all simulations. **a** Mean age of individuals—a measure of lifespan—measured at the end of the simulation ($t_{max}$); long lifespan is only found at high plasticity levels. **b** Mean evolved reproductive effort (gene value) covaries negatively with plasticity, measured as updating effort, $u$. **c** Mean population-wide mismatch against mean age of individuals measured at $t_{max}$. **d** The regression slope between individual mismatch and individual age, measured at $t_{max}$, with positive values indicating that older individuals are more mismatched to the current environment. Frequent enough plasticity can prevent this type of senescence. Symbol shapes indicate sampling error ($\varepsilon$); colour (from dark blue to yellow) indicates increasing environmental autocorrelation ($p$)

reproductive effort while they can retain variation in the precise level of plasticity (the evolved updating schedule $u$). Even so, the solutions clearly group into distinct categories of either high, low, or (in the case of highly autocorrelated environments) intermediate updating frequency (Fig. 2).

There are strong interactions between plasticity ($u$), reproductive effort $r$, and the consequent lifespan under different environments. Stable environments (high $p$) generally lead to less plasticity (yellow squares and diamonds, Fig. 2a, b), and this is associated with short lifespans (Fig. 2a) that reflect high reproductive effort (Fig. 2b). If these conditions combine with low updating error, reproductive effort evolves to be lower, yielding relatively long lifespans (yellow circles, Fig. 2a), while plasticity stabilizes at intermediate values (yellow circles, Fig. 2b). Here individuals are relatively well phenotypically matched to their environment, and long lifespan is only achieved by those not too mismatched to their environment (Fig. 2c, d).

Low updating error is clearly associated with 'slow' life histories (circles, Fig. 2a, b). The longest lifespans require both that updating error is low and that lifelong plasticity has evolved (in the form of frequent updating, i.e., high $u$, Fig. 2a). The effect of low updating error leading to a 'slow' life history is far clearer than low error conditions selecting for high updating frequency per se (Fig. 2a, b). Conversely, we find the fastest life histories when updates are error-prone (diamonds, Fig. 2a, b). The short lifespan of these cases is a consequence of little investment in plasticity, leading to eventual mismatches between individual traits and their current environment (Fig. 2c, d).

If environmental fluctuations are faster (green to blue points in all figures), the above result of high plasticity associating with a slow life history (Fig. 2a, b) remains robust, but the importance of the updates being able to reduce the mismatch without much error is even stronger than above. If updates lead to near zero mismatch, evolution proceeds towards lifelong plasticity (i.e., frequent updates), very low reproductive effort, and long lifespans (Fig. 2). Higher error (squares and diamonds in Fig. 2) clearly limits the lifespan. In combination with intermediate autocorrelation in the environmental fluctuations, individuals evolve low investment in plasticity and invest instead in fast reproduction (green to light blue squares and diamonds in Fig. 2a, b). In these populations, the levels of phenotypic mismatch are relatively high (Fig. 2c), and mismatch tends to increase with age (Fig. 2d). In populations with larger environmental fluctuations, stronger plasticity (more frequent updating) is again selected for; the relatively high level of mean phenotypic mismatch (due to updating error) does not, in this case, increase with age (dark blue squares and diamonds, Fig. 2d, see also Supplementary Fig. 1).

**Evolutionary dynamics**. We typically found no correlation between initial and evolved updating schedules ($u$), or between initial and evolved reproductive effort ($r$), for any species-specific parameter values; this indicates that the simulations proceeded beyond transient effects. However, a few exceptional cases indicate the possibility of alternative evolutionary equilibria. Low environmental autocorrelation and high updating error can yield two alternative life histories depending on the starting values used to initialize the population (Supplementary Figs. 1 and 3). The usual pattern here is repeated updating (high $u$) and a slow life history, but in some cases, the population instead evolves a mirror image life history with little plasticity and high reproductive effort (Fig. 2b) that combines with strong apparent senescence due to increase in phenotypic mismatch with age (Fig. 2d).

It is also interesting to note that the modelled traits evolve at very different rates. From randomly chosen initial values, reproductive effort evolves very fast to match an optimum that

is valid for the current (i.e., initial) updating schedule (near-vertical lines in Supplementary Fig. 2). Plasticity itself evolves much more slowly, with reproductive effort quickly following any (slow) changes in the updating frequency. Observing this process over time leads to a tight negative correlation between reproductive effort and plasticity in most populations. Exceptions occur in populations with both high autocorrelation and high update error; here a constant, intermediate reproductive effort is observed, with plasticity evolving to be low and, due to environments being relatively stable, not impacting the life history much (Supplementary Fig. 2). In other words, life histories are not expected to respond strongly to reversible plasticity when the environment changes only slowly and when phenotypic updates are not very successful in reducing the mismatch between individual traits and current environmental conditions.

**Model robustness**. We found the model to be highly robust to changes in the costs of sampling and plasticity. We ran the model with both higher and lower fecundity costs of phenotypic updating ($\kappa = 0.2$ and $0.6$), and saw no qualitative changes in the results; quantitatively, we observed more frequent updating when the costs of doing so were lower (Supplementary Note 1; Supplementary Figs. 4 and 5). With increased costs of reproduction ($\rho = 0.05$), we find that populations either evolve similar strategies as with lower costs or alternatively so high reproductive efforts that all individuals die and there is no selection for plasticity (Supplementary Note 2; Supplementary Fig. 6). We also let the phenotypic mismatch affect adult mortality only, with no effect on clutch size, and again found no qualitative change in the results (see Supplementary Note 3; Supplementary Fig. 7). In a further model variant where phenotypic mismatch did not affect adult mortality, but the fecundity effect was assumed present as in the main model, we observed a shift in the results. In this case, reproductive effort evolved to be nearly constant and independent of reversible plasticity. The same lack of association occurred between lifespan and plasticity (see Supplementary Note 4; Supplementary Fig. 8), with lifespan evolving to a similar value across all updating schedules. This set of additional results enables us to ask what drives the results in the main model, with two alternative interpretations: lifespan differences may reflect coevolution between reversible plasticity and reproductive effort (with lifespan trade-offs), or reduced lifespan in populations with rare phenotypic updating could reflect poor average match with the environment. To disentangle these potentially co-occurring factors, it is useful to graph the relationship between mortality and reversible plasticity; here, phenotypic mismatch and mortality costs of reproductive effort were found to be of comparable size (Fig. 3). From this, we conclude that even if the assumed relationship between phenotypic mismatch and mortality affects lifespan in a way that necessarily makes plasticity covary with lifespan, the co-evolved reproductive effort contributes equally to the reduction in lifespan in this model.

Because changing $p$ in the main model does not only alter the environmental autocorrelation, but also the total environmental variance, we also rerun the model with a more complex simulation of the environment (an ARMA(2,1) model) to confirm that it is indeed the environmental autocorrelation and not the total environmental variance that affects which strategies evolve (see Supplementary Note 5 and Supplementary Figs. 9–11). These simulations show very similar results to the main model, highlighting the crucial role of the environmental autocorrelation driving the results.

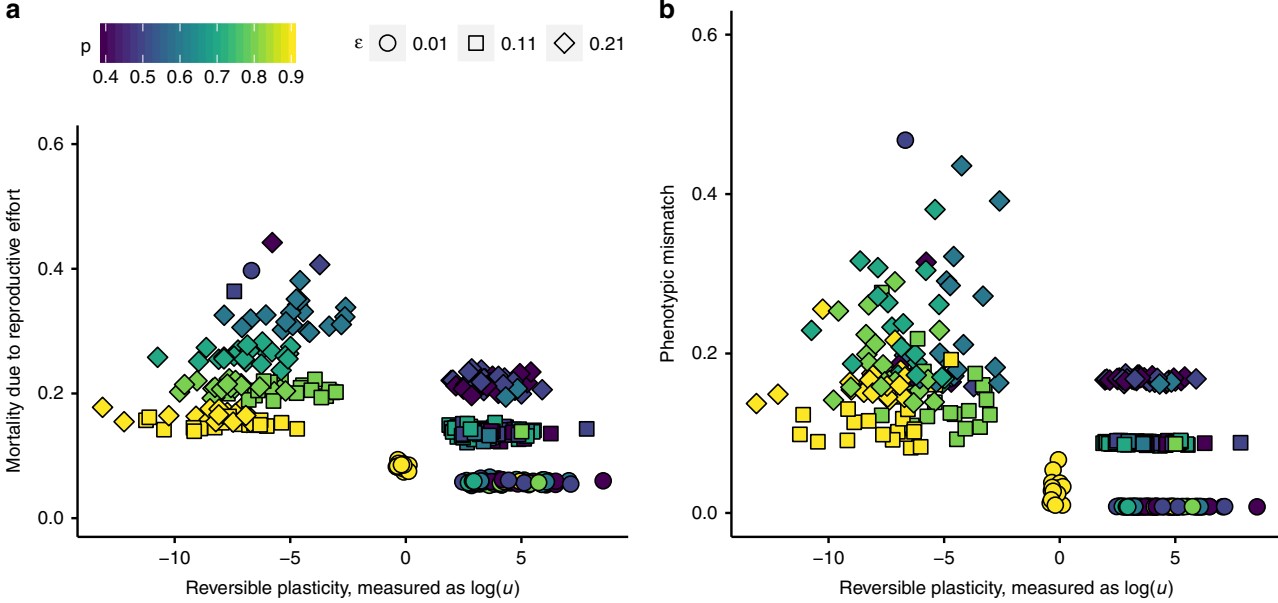

**Fig. 3** Different sources of mortality. **a** Mortality due to mean reproductive effort in all populations measured at the end of the simulation ($t_{max}$); there is a negative correlation because reproductive effort co-evolved to be negatively correlated to reversible plasticity (see Fig. 1b). **b** Mean population-wide mismatch against plasticity, measured as updating effort, $u$ recorded at $t_{max}$. Symbol shapes indicate updating error, $\varepsilon$; colour (from dark blue to yellow) indicates increasing environmental autocorrelation ($p$)

## Discussion

Our model forms a clear proof of principle that a long life is not merely a context that sets the stage for lifelong plasticity, but that plasticity itself can select for a longer life. This changes the per-spective on how causalities should be understood in cases where reversible plasticity is found to covary with lifespan. Previous models predict that investment in plasticity should only evolve when the environment varies at a suitable rate with respect to the individual's life history[3,17,19]. Our results show this to be only one side of the coin: plasticity itself can shift the balance in well-known trade-offs between survival and reproductive effort to favour the parent's own survival.

We also show some counterintuitive patterns. Slow environ-mental fluctuations, making any benefits of plasticity small, select against reversible plasticity, and this can lead to phenotypes that are poorly matched to the environment as a whole. In these highly autocorrelated environments, a relatively fast life history proves to be a better solution than any attempt to keep an individual's own survival intact via plasticity. In an avian context, Sol et al.[13] point out that innovation propensity appears to pre-dict maximum lifespan, and part of this (but not all) comes from a joint association with brain size, which could indicate high investment in behavioural flexibility.

The fast–slow life history continuum has been investigated across a range of species[21–24], but we have not found examples of plasticity included in these studies. On the other hand, meta-analytic techniques have been used to quantify plasticity in a standardized manner across species[25,26], but here, the studies have not been interested in lifespan variation. Clearly, there is scope for both methods to be employed in the same studies, as this would allow interspecific tests of our predictions; local adaptation can also be used as a framework permitting multi-population studies within a single species[27]. Alternatively, as a stronger method in terms of detecting causalities, experimental evolution approaches could investigate the coevolutionary pat-terns of lifespan and plasticity when environments vary in their autocorrelation structure. Experimental evolution approaches have already been used to select for plasticity[28] on the one hand

and senescence[29] on the other; a simultaneous investigation of both responses could prove fruitful, and this approach could be made even more powerful by posing lifespan limitations on some lines but not others. A third approach could specifically focus on learning as a specific form of behavioural plasticity, and fourth, differences between the sexes in plasticity[30,31] and lifespan[29] could—again, if combined in a single study—also inform the validity of our model (though here we note that our model, being a first step in the exploration of coevolutionary effects, ignored all complications of sexual reproduction).

Previous models have shown the potential for older individuals to become more mismatched to their environment if the envir-onment is changing over time[20]. We show that this process of becoming outdated can occur within a species, while the opposite pattern (long life associating with low mismatch) can be predicted in between-population or between-species comparisons. In the latter kind of comparison, environments are likely to differ between species, and investment in reversible plasticity can lead to long lifespans that feature good matches between phenotypes and genotypes. We predict that the populations that are most mismatched to their environments live in environments that have intermediate autocorrelation, and in which accurate updating is not possible (i.e., updating is an error-prone process that can even increase the mismatch). Under such conditions, investing in reversible plasticity is selected against, and individuals conse-quently become more phenotypically mismatched after one sea-son than individuals that live and evolve in more stable environments. The life history can still permit annual survival rates high enough to allow many individuals to survive beyond one season. The populations with the strongest increase in phe-notypic mismatch with age are those that live in fast changing environments in which updating is very error-prone, effectively preventing plasticity as a solution to rapid environmental changes.

In conclusion, our study promotes the view that species can be placed along a fast–slow continuum of life history patterns in a manner that covaries with the degree of plasticity, with the causalities forming a feedback loop rather than being

unidirectional. Although disentangling coevolving effects can be empirically difficult, we urge investigators to not assume a priori that the type of lifespan evolves first and plasticity follows; indeed, we found reproductive effort to respond very fast to environmental sampling regimes (plasticity), with the reverse rate of evolution being much slower.

## Methods

**Model description**. We use an individual-based simulation model where we model genotypes, phenotypes, survival, and reproduction individually for all members of the population. Population size is constantly 2000 individuals. An individual life lasts, potentially, several time steps, and the environment changes from one time step to the next. Since an individual is allowed to reproduce in every time step, generations are overlapping.

The environment, $E$, varies on a continuous scale, and in each time step, $t$, the environment has one component that is drawn from the standard normal distribution and one component from the previous environment. The relative weight of these two components is governed by a parameter, $p$, which therefore determines the autocorrelation in the environment: $E_t = p\,E_{t-1} + (1-p)\,\zeta$, where $\zeta$ is independently drawn from $N(0,1)$. The value of $E_t$ is interpreted as the phenotypic value that results in zero mismatch; all other phenotypes have positive mismatch $m_{i,t} = |x_{i,t} - E_t|$, where $x_{i,t}$ is the individual's phenotype. Larger mismatch values translate into poorer reproduction and survival, as explained below. For illustration purposes, we also present statistics on the increase in phenotypic mismatch with age. This is done by extracting the slope from a linear regression of $m_{i,t} \sim \text{age}_{i,t}$ including all individuals present in a population for the last 10 time steps of the simulation, to average out effects of any particular environment or environmental change.

An individual's genotype consists of two components (loci) that are passed on to the offspring (with some mutation as described below): plasticity, which we model as the updating schedule $u$, and the reproductive effort, $r$. Each individual, at the beginning of each time step, either updates or does not update its phenotype, with the probability that updating occurs modelled as an increasing function of the time $T_u$ since the last update. The genotype, $u$, determines the how steeply the probability rises with $T_u$ (high $u$ implies frequent updating): $P[\text{updating}]_i = 1 - e^{-u_i T_{u,i}}$, where the subscript $i$ refers to the individual in question. We assume some plasticity always occurs, in that newborn individuals (whose $T_{u,i}$ value are necessarily undefined) perform an update with probability 1 regardless of $u_i$. Updating does not completely reset mismatch to zero, instead, updating leading to phenotype that includes an error term: the post-update phenotype is $E_t + N(0,\varepsilon)$. If $\varepsilon = 0$, updates will always perfectly remove the mismatch, while large $\varepsilon$ can in principle also increase the mismatch between the organism and the environment at an update event. Individuals that did not perform an update are assumed keep their phenotype from the previous time step; $x_{i,t} = x_{i,t-1}$.

Each time step, updating is followed by reproduction. Reproduction is asexual. The clutch size, $c_{i,t}$, produced by each individual depends on three factors. The first, denoted $r_i$, is a component of the genotype that denotes reproductive effort, bounded by an upper limit of 10. The second component is the level of mismatch to the current environment ($m_{i,t}$), and the third component is a penalty ($\kappa$) for individuals that in this time step performed an update. Expected clutch size combines these effects as

$$c_{i,t} = \min(10, r_i)\, e^{-m_{i,t}^2}(1 - \kappa\, U_{i,t}),$$

where $U$ takes the value 1 for individuals who updated, and 0 otherwise (Fig. 1b). Where not otherwise indicated, we use a baseline updating cost of $\kappa = 0.4$.

Each individual's genotype values $u_i$ (for sampling effort) and $r_i$ (for reproductive effort) are inherited by all its offspring, with each of them also independently experiencing mutation, to yield $u_j = e^{\log(u_i)+\mu_u}$ and $r_j = e^{\log(r_i)+\mu_r}$, where each $\mu$ is drawn from the normal distribution, $\mu \sim N(0, 0.01)$.

Parents die with probability $P[\text{death}]_i = \alpha_0 + m_{i,t} + \rho r_i^2$ (Fig. 1c), where $\alpha_0 = 0.05$ is a baseline mortality, to which we add the mismatch with the environment and a term reflecting the cost of reproduction (high $\rho$ implying large cost), which increases with an individual's reproductive effort $r_i$. New recruits replace dead adults, until there are no further vacancies in the population. Parents that are allowed to provide a recruit are drawn randomly from the population but probability of providing a recruit is weighted by expected clutch size $c_{i,t}$.

The simulations are allowed to continue for $t_{\max} = 50{,}000$ time steps, At this point, we record the age and genotypic distribution of all individuals. In most cases, genotype values stabilize long before this, but in a few cases evolution needs a long time to reach the final values. Fitness components (survival, lifespan, and clutch size of each individual) are emergent properties of the model.

We vary both the autocorrelation in the environment, $p$, and the updating error, $\varepsilon$. In each of these environmental scenarios, we start the simulation from a range of genetic values for $u_{\text{init}} = \{0.0100, 0.0316, 0.1000, 0.3162, 1.0000, 3.1623, 10.000\}$ and for three different values of $r_{\text{init}} = \{1, 4, 7\}$.

Figures are produced with R, version 3.5.0 and the package 'ggplot2'[32], 'viridis'[33] and 'gridExtra'[34].

**Code availability**. The model is coded in MATLAB and code can be found at Dryad Digital Repository, https://doi.org/10.5061/dryad.m7b43mm.

## Data availability

Data from simulation output is available in Dryad Digital Repository, https://doi.org/10.5061/dryad.m7b43mm.

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

## Acknowledgements

I.I.R. was supported by the Norwegian Research Council (project number 240008 to I.I.R., and partly through its Centres of Excellence funding scheme, project number 223257) and by NTNU. H.K. acknowledges support by the SNSF.

## Author contributions

H.K. conceived the initial idea and both authors developed the idea in collaboration. Both authors contributed to the coding of the model. I.I.R. ran simulations and visualized results. Both authors wrote the manuscript in collaboration.

## Additional information

**Competing interests:** The authors declare no competing interests.

