## [Peer Review File · Nature Communications]

Reviewers' Comments:

Reviewer #1:

Remarks to the Author:

In this manuscript, Ratikainen and Kokko develop a model to explore the co-evolution of phenotypic plasticity (sampling), and reproductive investment (with an implicit trade-off between current and future reproduction). They show that fluctuating environments (with low autocorrelation) select for higher sampling (i.e., phenotypic plasticity), which in turn favours low investment in current reproduction, resulting in increased survival. Hence, selection for higher plasticity results in longer lifespans. This is a novel result, and is contra to the common assumption that long lifespans are required first in order for selection to favour higher levels of plasticity. I think this model provides an important contribution that will be of broad interest, including to researchers interested in the life-history evolution, sampling behaviour, and phenotypic plasticity. I have a few general comments that I hope could be useful in strengthening this contribution.

First, in the introduction, age-dependent plasticity, age-dependent expectation of phenotypic mismatch, and senescence are all referenced in developing the rationale for this study. These are all within-individual processes, yet all the results (including in the appendix) are presented as population means. Providing figures that illustrate the within-individual processes would be a meaningful addition, and would create greater coherence between the introduction and results.

Second, I think it would be nice to know whether some of the model assumptions affect the outcomes. Specifically, how robust are the results to variation in the cost of sampling (cs)? Also, the model assumes that phenotypic mismatch confers both a survival and reproduction cost. Do the results change if the cost is on one or the other?

Finally, in the discussion, you write that the lack of empirical tests might be due to the difficulty in placing species along a unidimensional axis of plasticity, but that such quantifications would be an interesting avenue for future research (lines 130-133). Please give concrete suggestions here. What type of systems, what type of traits, would you recommend for future empirical work? What are the key assumptions of your model that such systems should fulfill? If this model doesn't lend well to empirical tests, and is meant to serve as an untestable proof of concept, that's also fine in my opinion, as it's still illuminating as a model. But this should be made explicit to avoid future empirical tests being purported to test this model when in fact they don't (e.g., because they violate critical assumptions).

Minor comments:

Line 15: "reversible phenotypic plasticity... is expected to evolve if environments vary relatively predictably within....". Please rephrase to make clear that what is predictable is that there is variance in the environmental conditions, but that the specific state of the environment is not predictable (and hence requires sampling).

Line 40: how do you combine insights from age-dependent plasticity? You do not model age-dependent effects.

Line 117: use of "merely" and "simply" in same sentence is redundant. Delete one.

Lines 224-225: "the maximum potential fitness is equal for all environments, and there are therefore no "good" or "bad" environments". If the probability of achieving maximum fitness differs, then I would consider that there are "good" and "bad" environments. Please rephrase.

Line 228: insert "the", should be "...until the next..."

Lines 237-238: description doesn't match equation. $(age_i - u_i) = \text{current age} - \text{age when last sampling event occurred}$ (i.e., time elapsed since last sampling). Please rephrase to clarify.

Figure A1: define p and ϵ in figure legend.

Figure A2: Correct type (should be "Within population....", also, define p and ϵ in figure legend.

Figure A3: colours described in figure axes and inset figure legend don't match text description in written figure legend. I recommend splitting this into two separate panels, grouping specific model runs (black and dark red, versus grey and red).

Reviewer #2:

Remarks to the Author:

I started reading this paper with the sense that I would learn things that were really interesting and novel. Unfortunately I found the paper confusing so it is not clear to me that any such insights or

conclusions are to be found here. It is possible that with a significant rewrite the message here can be made sufficiently clear but I am not sure that this is the case. Here are my criticisms.

1. Given that the paper is singularly devoid of any mathematics, it is not clear to me what assumptions matter to the model. At the simplest level, I could argue that the parameter s that is identified here as plasticity (and is actually some kind of sampling) determines the mismatch, and the paper assumes that mortality increases with mismatch. My conclusion is that increasing s will decrease mismatch and so mortality and so lengthen life.

What the paper actually does is to make assumptions about (a) The environment and its autocorrelation -- it is not clear to me that the authors allow for the fact that changing autocorrelation automatically changes the long term environmental variance. It is also not clear to me what drives the relationship between autocorrelation and lifespan. (b) There are in fact no genes here: everything is phenotypic, so the sudden appearance of genes in the specification of the model is, at the least, not useful. (c) What does finite population size have to do with the results? Would you not get exactly the same thing with a population that is growing in size? (d) I know that it is popular to describe fitness as a property of individuals, but here it is a property of the phenotypic characteristics.

2. The model is actually simple and your discussion would make a great deal more sense if you presented the model first. In my opinion an appendix is useful only when it contains developments that can be effectively summarized in the main text. That is not the case here. Without some explicit development of the quantitative underpinnings, I found the discussion extremely confusing.

Response to reviewers

Reviewer #1 (Remarks to the Author):

In this manuscript, Ratikainen and Kokko develop a model to explore the co-evolution of phenotypic plasticity (sampling), and reproductive investment (with an implicit trade-off between current and future reproduction). They show that fluctuating environments (with low autocorrelation) select for higher sampling (i.e., phenotypic plasticity), which in turn favours low investment in current reproduction, resulting in increased survival. Hence, selection for higher plasticity results in longer lifespans. This is a novel result, and is contra to the common assumption that long lifespans are required first in order for selection to favour higher levels of plasticity. I think this model provides an important contribution that will be of broad interest, including to researchers interested in the life-history evolution, sampling behaviour, and phenotypic plasticity. I have a few general comments that I hope could be useful in strengthening this contribution.

Thank you!

First, in the introduction, age-dependent plasticity, age-dependent expectation of phenotypic mismatch, and senescence are all referenced in developing the rationale for this study. These are all within-individual processes, yet all the results (including in the appendix) are presented as population means. Providing figures that illustrate the within-individual processes would be a meaningful addition, and would create greater coherence between the introduction and results.

This is a good suggestion. We have combined it with a need to explain the model assumptions more clearly (reviewer #2). Our new figure 1 shows all the main individual level trade-offs from the model, while the other figures show population-level means for a reason: many of our predictions are of an interspecific nature, such as “how does the evolution of plasticity impact lifespan”?

Second, I think it would be nice to know whether some of the model assumptions affect the outcomes. Specifically, how robust are the results to variation in the cost of sampling (c_s)? Also, the model assumes that phenotypic mismatch confers both a survival and reproduction cost. Do the results change if the cost is on one or the other?

We have now run various modified versions of the model to answer these questions, finding that the results are generally very robust to these assumptions. These additional results are now described in the new section “Model robustness” and in the new ESM.

Finally, in the discussion, you write that the lack of empirical tests might be due to the difficulty in placing species along a unidimensional axis of plasticity, but that such quantifications would be an interesting avenue for future research (lines 130-133). Please give concrete suggestions here. What type of systems, what type of traits, would you recommend for future empirical work? What are the key assumptions of your model that such systems should fulfill? If this model doesn't lend well to empirical tests, and is meant to serve as an untestable proof of concept, that's also fine in my opinion, as it's still illuminating as a model. But this should be made explicit to avoid future empirical tests being purported to test this model when in fact they don't (e.g., because they violate critical assumptions).

We agree: it is more helpful to provide concrete suggestions than mere hand-waving. Although we are constrained by the need to keep our text concise, we now write (lines 290-294, in track-changes version of ms):

“A possible way forward could be to investigate a single very important trait that varies in the degree of plasticity across populations or species. Alternatively, experimental evolution approaches could investigate the coevolutionary patterns of lifespan and plasticity when environments vary in their autocorrelation structure; this approach could be made more powerful by posing lifespan limitations on some lines but not others.”

Minor comments:

Line 15: “reversible phenotypic plasticity... is expected to evolve if environments vary relatively predictably within.....”. Please rephrase to make clear that what is predicatable is that there is variance in the environmental conditions, but that the specific state of the environment is not predictable (and hence requires sampling).

We have rephrased the sentence to clarify; also we now refrain from making an “will always evolve” type prediction, as we merely mean that these are the conditions when one can expect there to be selection for plasticity (lines 19-20):

“Reversible phenotypic plasticity, the ability to change ones phenotype repeatedly throughout life, can be selected for in environments that do not stay constant throughout an individual’s lifetime.”

Line 40: how do you combine insights from age-dependent plasticity? You do not model age-dependent effects.

We have rephrased — ‘complement’ is a better word to describe what we do than ‘combine’, as we indeed do not model sampling effort as a function of age. In other words, we view our model as complementary to others’ efforts, who have different research foci.

Line 117: use of “merely” and “simply” in same sentence is redundant. Delete one.

Thank you, we have deleted “simply”.

Lines 224-225: “the maximum potential fitness is equal for all environments, and there are therefore no “good” or “bad” environments”. If the probability of achieving maximum fitness differs, then I would consider that there are “good” and “bad” environments. Please rephrase.

We agree, and believe this sentence offered potential to lead to misunderstandings more than it helped to understand that is going on. We have simply deleted the sentence.

Line 228: insert “the”, should be “...until the next...”

Thank you, now corrected.

Lines 237-238: description doesn’t match equation. $(age_i - u_i)$ = current age minus age when last sampling event occurred? (i.e., time elapsed since last sampling). Please rephrase to clarify.

There was a typo in the equation, and we’re grateful that the reviewer had such sharp eyes. This has been corrected now. Thank you for spotting this.

Figure A1: define p and ϵ in figure legend.

Figure A2: Correct type (should be “Within population....”, also, define p and ϵ in figure legend.

Thank you. The typo is corrected, and p and ϵ are now defined in both figure legends.

Figure A3: colours described in figure axes and inset figure legend don't match text description in written figure legend. I recommend splitting this into two separate panels, grouping specific model runs (black and dark red, versus grey and red).

Thank you! We have now split this into two separate figures as suggested, and this should also clarify any confusing legends.

Reviewer #2 (Remarks to the Author):

I started reading this paper with the sense that I would learn things that were really interesting and novel. Unfortunately I found the paper confusing so it is not clear to me that any such insights or conclusions are to be found here. It is possible that with a significant rewrite the message here can be made sufficiently clear but I am not sure that this is the case. Here are my criticisms.

1. Given that the paper is singularly devoid of any mathematics, it is not clear to me what assumptions matter to the model. At the simplest level, I could argue that the parameter s that is identified here as plasticity (and is actually some kind of sampling) determines the mismatch, and the paper assumes that mortality increases with mismatch. My conclusion is that increasing s will decrease mismatch and so mortality and so lengthen life.

This is correct. That said, we are not entirely sure how to interpret the reviewer's tone when commenting on the relationship between plasticity and sampling ('actually some kind of sampling'). Plasticity cannot occur without sampling the environment — except if one wishes to use it as an umbrella term that also includes phenotype switching (which can occur and be selected for without the organism measuring the state of the environment in any way, e.g. Beaumont et al. 2009 Nature). Our aim is to consider the more usual case of plasticity which necessitates that the organism measures the state of the environment. We now clarify this:

“Phenotypic plasticity is defined as the ability of one genotype to produce more than one phenotype depending on some environmental variable¹ (in this view, which we follow here, plastic traits necessarily involve sampling the environmental state at least once, and our work does not consider the alternative of phenotypic switching² that can occur without any sampling effort by the organism).”

We would also like to remark that while s on its own lengthens life, mortality is also impacted by reproductive effort, which in turn evolves to correlate negatively with plasticity. This, too, contributes to the longer lifespans in populations where individuals are more plastic. We have now elaborated on this in the new section “Model robustness” and with the new figure 3.

We have also moved the model description into the “results” section after advice from the editor, and all the equations are now easily available.

What the paper actually does is to make assumptions about (a) The environment and its autocorrelation -- it is not clear to me that the authors allow for the fact that changing autocorrelation automatically changes the long term environmental variance. It is also not clear to me what drives the relationship between autocorrelation and lifespan.

The reviewer identifies a real concern here. Changing the environmental parameter p (Figure 1) has an effect on the autocorrelation (our intended effect), but also on the total environmental variation, and it is not possible for us to state which is responsible for the evolutionary outcomes

we observe. We have therefore added a new set of results: we now also run the model with an environment that fluctuates according to an ARMA(2,1) model, allowing us to choose parameter such that the correlation between time steps varies while the long term environmental variance is nearly constant. The results are very similar to our original results, both qualitatively and quantitatively (see ESM). We now report these additional results in a separate “model robustness” section, described in detail in Electronic Supplementary Material.

The relationship between autocorrelation and lifespan can be explained as follows. A low-autocorrelation environment presents survival challenges that are fundamentally more difficult to deal with than high-autocorrelation environments: the same phenotype that did well in year 1 will not do well in year 2 if the environment is now dramatically different. Plasticity (via sampling the new environmental state) can in principle mitigate this effect, but plasticity has its own costs (we assume a cost on fecundity) and the environmental state is unlikely to be measured perfectly. Contrast this with a perfectly autocorrelated, i.e. constant, environment, where the mismatch can between the phenotype and the environment is expected to disappear over evolutionary time, without the need to employ plasticity at all.

(b) There are in fact no genes here: everything is phenotypic, so the sudden appearance of genes in the specification of the model is, at the least, not useful.

We are dealing with a situation where we address a phenomenon in the absence of knowledge of any specific genetic architecture. This means that it is difficult to win an argument over the best level of detail in the genotype-phenotype map. However, we politely disagree that talking about genes / genotypes is not useful. The goal is to see which phenotypes evolve, in the sense of making predictions about observable life-history traits such as lifespan or reproductive effort. But we also need to make a distinction between what can be observed directly and what is assumed to be inherited from parent to offspring. It would clearly be nonsensical (one could use the word Lamarckian) to assume that all components of the phenotype (e.g. the axis labels of the various figures presented) are inherited directly. Instead, clarity is maximized when we use genotypes (previously ‘genes’) to refer to those aspects that are passed on to offspring. Note that our terminology follows the gist of e.g. Edelaar et al. 2017, an evolutionary model of phenotypic plasticity, although we avoid some of their complexities such as circular trait spaces.

Edelaar, P., Jovani, R. & Gomez-Mestre, I. 2017. Should I change or should I go? Phenotypic plasticity and matching habitat choice in the adaptation to environmental heterogeneity. Am. Nat. 190:506-520.

We have rewritten parts of the model description to make our approach clearer.

(c) What does finite population size have to do with the results? Would you not get exactly the same thing with a population that is growing in size?

Real populations experience population regulation, and unlimited population growth is in the long term not observed. This is the first reason to consider evolution with finite population size. The 2nd reason is a practical one. Modelling approaches differ in whether they’re easier to work with when ignoring density regulation or when assuming such an effect is in place, and this typically has a strong effect on the emphasis placed in the model. Lion 2018 has provided a fantastic explanation of how classical population genetics models usually ignore leave population regulation unconsidered. In simulation approaches, the convenience factor works the other way: when all individuals need to be tracked, the problem becomes computationally easier if unlimited

population growth is prevented. As computational efficiency matches biological reality, we see no reason to change our modelling choice regarding this matter.

Lion, S. 2018. Theoretical approaches in evolutionary ecology: environmental feedback as a unifying perspective. Am. Nat. 191:21-44.

(d) I know that it is popular to describe fitness as a property of individuals, but here it is a property of the phenotypic characteristics.

We use the word 'fitness' in contexts where it gives intuitive insight into the reasons why certain traits spread in the population (long life and good reproductive success both make an individual more fit). The modelling work itself does not use the concept: we are not assigning fitness values to individuals to then decide on the parents of the next generation. It is true that we are following a popular route here, but we do not really see a reason to avoid doing so, as it helps heuristically to see what is going on.

2. The model is actually simple and your discussion would make a great deal more sense if you presented the model first. In my opinion an appendix is useful only when it contains developments that can be effectively summarized in the main text. That is not the case here. Without some explicit development of the quantitative underpinnings, I found the discussion extremely confusing.

Yes, we agree with this and we have now described the model in the results section.

Reviewers' Comments:

Reviewer #1:

Remarks to the Author:

I think that the authors have addressed most of the points raised in the initial review through their revisions. One point that I don't think was addressed very satisfactorily was my earlier point about recommendations for empirical systems in which to test the model. The text added (lines 290-294) proposed measuring a single "very important" trait. This is rather vague. What constitutes "a very important" trait? How often would there actually be sufficient data on plasticity for the same trait across multiple distinct populations? If the authors don't know of any systems that are well suited to test the model, providing a small checklist of requirements would be useful to allow empiricists to assess whether their system is tractable or not. Other than this, I have only a few very minor comments stemming from the revision.

1. I don't agree with the statements that plasticity traits necessarily involve sampling the environmental state at last one. I think of sampling as a decision to reduce uncertainty about the environment (i.e., gather information). However, many of the classic examples of phenotypic plasticity are in response to environmental states that organisms do not need to sample in order for their phenotype to be affected (e.g., extreme temperatures). I think it's worthwhile to distinguish between active sampling and passive exposure to cues.

2. The section on model robustness is not as polished as the rest of the text (e.g., changing verb tenses). This should be tightened up, however, the content of the text additions is excellent and significantly improves the MS.

3. Typo in Figure Legend A1 and A2.. Lines 330 and 338 should read "sampling" (not "samling").

Reviewer #2:

None

Response to referee

Here we describe our responses to the referee and our changes in the ms. Line numbers refer to the line numbers in the “simple markup” tracked changes in the main ms., unless otherwise indicated. We have also coloured our changes in yellow to make them easier to find in the track-changes version of the ms.

REVIEWERS' COMMENTS:

Reviewer #1 (Remarks to the Author):

I think that the authors have addressed most of the points raised in the initial review through their revisions.

Thank you for your assessment!

One point that I don't think was addressed very satisfactorily was my earlier point about recommendations for empirical systems in which to test the model. The text added (lines 290-294) proposed measuring a single “very important” trait. This is rather vague. What constitutes “a very important” trait? How often would there actually be sufficient data on plasticity for the same trait across multiple distinct populations? If the authors don't know of any systems that are well suited to test the model, providing a small checklist of requirements would be useful to allow empiricists to assess whether their system is tractable or not.

Thank you for forcing us to look more carefully at the literature – we indeed found ways to improve the text further. We have identified several new studies (see lines 186-199) that provide, hopefully, much inspiration for further work in this area; basically, we're dealing with a situation where studies have used certain methods for plasticity research, other studies have developed other methods for lifespan (or fast/slow life history) research, and these could be very usefully combined.

Other than this, I have only a few very minor comments stemming from the revision.

1. I don't agree with the statements that plasticity traits necessarily involve sampling the environmental state at least once. I think of sampling as a decision to reduce uncertainty about the environment (i.e., gather information). However, many of the classic examples of phenotypic plasticity are in response to environmental states that organisms do not need to sample in order for their phenotype to be affected (e.g., extreme temperatures). I think it's worthwhile to distinguish between active sampling and passive exposure to cues.

The sentence in question, at the very beginning of the Introduction, was intended to distinguish between scenarios where the organisms responds to an environmental state and those in which the organism switches phenotype irrespective of the state of the environment. Thus the key is whether some kind of measurement of the environment is occurring or not; our emphasis “at least once” was misleading and we have now changed this so that any kind of causality from environment to phenotype is included:

“Phenotypic plasticity is defined as the ability of one genotype to produce more than one phenotype depending on some environmental variable¹. This definition differs from phenotypic switching², a phenomenon where phenotypic change occurs without being related to the state of the environment as experienced by the organism.”

We agree that ‘sampling’ as a term is perhaps too narrow. We are interested in any kind of updating event where the phenotype is adjusted to the demands of the current environment. We therefore now refer to updating effort rather than sampling effort, and have changed the notation such that the variable name s is now u . This has the added advantage that the new variable name (u) can no longer possibly be confused with survival. We have rewritten the entire MS so that we now refer to updating rather than

to sampling, which indeed makes it visible that the results are broadly applicable. (It was, in any case, assumed that the entire phenotypic updating is costly, and the 'sampling' was only the first step of this process. It is in hindsight better to avoid the word 'sampling' altogether.)

2. The section on model robustness is not as polished as the rest of the text (e.g., changing verb tenses). This should be tightened up, however, the content of the text additions is excellent and significantly improves the MS.

Thank you. We have checked this text with a particularly fine comb and made several changes in this section. Regarding tenses, we use the convention that results that "always hold" if true, e.g. something drives something in a model (in general), are in present tense; while one-off results (when we varied this, we found that) are in past tense. There were indeed some inconsistencies in the previous version and we have now corrected these.

3. Typo in Figure Legend A1 and A2.. Lines 330 and 338 should read "sampling" (not "samling").

Thank you. As we have now changed sampling to updating throughout, this is no longer a problem.